# Genotyping-by-Sequencing Derived Genetic Linkage Map and Quantitative Trait Loci for Sugar Content in Onion (*Allium cepa* L.)

**DOI:** 10.3390/plants10112267

**Published:** 2021-10-22

**Authors:** Ye-Rin Lee, Cheol Woo Kim, JiWon Han, Hyun Jin Choi, Koeun Han, Eun Su Lee, Do-Sun Kim, Jundae Lee, Muhammad Irfan Siddique, Hye-Eun Lee

**Affiliations:** 1Vegetable Research Division, National Institute of Horticultural and Herbal Science, Rural Development Administration, Wanju 55365, Korea; lyr1219@korea.kr (Y.-R.L.); cwkim@korea.kr (C.W.K.); support06@korea.kr (J.H.); hke1221@korea.kr (K.H.); lus4434@korea.kr (E.S.L.); greenever@korea.kr (D.-S.K.); arafay68@yahoo.com (M.I.S.); 2Postharvest Research Division, National Institute of Horticultural and Herbal Science, Rural Development Administration, Wanju 55365, Korea; chyunjin@korea.kr; 3Department of Horticulture, Institute of Agricultural Science & Technology, Jeonbuk National University, Jeonju 54896, Korea; ajfall@jbnu.ac.kr

**Keywords:** sugar content, GBS, genetic linkage map, QTL mapping

## Abstract

Onion (2n = 2x = 16) has been a nutritional, medicinal and economically valuable vegetable crop all over the world since ancient times. To accelerate the molecular breeding in onion, genetic linkage maps are prerequisite. However, construction of genetic linkage maps of onion remains relatively rudimentary due to a large genome (about 16.3 Gbp) as well as biennial life cycle, cross-pollinated nature, and high inbreeding depression. In this study, we constructed single nucleotide polymorphism (SNP)-based genetic linkage map of onion in an F_2_ segregating population derived from a cross between the doubled haploid line ‘16P118’ and inbred line ‘Sweet Green’ through genotyping by sequencing (GBS). A total of 207.3 Gbp of raw sequences were generated using an Illumina HiSeq X system, and 24,341 SNPs were identified with the criteria based on three minimum depths, lower than 30% missing rate, and more than 5% minor allele frequency. As a result, an onion genetic linkage map consisting of 216 GBS-based SNPs were constructed comprising eight linkage groups spanning a genetic length of 827.0 cM. Furthermore, we identified the quantitative trait loci (QTLs) for the sucrose, glucose, fructose, and total sugar content across the onion genome. We identified a total of four QTLs associated with sucrose (*qSC4.1*), glucose (*qGC5.1*), fructose (*qFC5.1*), and total sugar content (*qTSC5.1*) explaining the phenotypic variation (R^2^%) ranging from 6.07–11.47%. This map and QTL information will contribute to develop the molecular markers to breed the cultivars with high sugar content in onion.

## 1. Introduction

Onion (*Allium cepa* L.; 2n = 2x = 16) is one of the most widely cultivated and economically and nutritionally important vegetable crops worldwide. Aside from the culinary uses, onion has several medicinal and nutritional benefits [1,2]. The global production value of onion stands second after tomatoes [3].

The quality of fruits and vegetables is largely attributed to the sweetness determined by the level of soluble sugar content. In both vegetables and fruits, sweetness results from sugars such as sucrose, glucose and fructose [4], with fructose being the sweetest, followed by sucrose and glucose [5]. Sugar content not only affects the sweetness of vegetables and fruits, but also influences the taste and palatability [6]. The onion bulbs contain sucrose, glucose, and fructose as the free sugar components which comprise the majority (65–80%) of onion bulb dry matter content [7]. The sugar content affects the sweetness of the bulbous onion as well as ascertains the sugar to spiciness ratio which determines the overall onion flavor [8]. In addition, the onion bulb stores the formed fructose polysaccharides, which also provides tolerance against cold and drought stress [9].

Genomic and genetic studies of onion have been limited because of its enormous genome size (16.3 Gbp), cross-pollinating nature, biennial life cycle, high level of heterozygosity, inbreeding depression, and laborious breeding system [1,10]. This is why the high-quality whole genome sequencing of onion has not been completed yet. However, a recent study that describes the de novo assembly and created eight pseudomolecules with five genetic linkage maps using the doubled haploid onion line “DHCU066619” is the latest high-quality onion reference genome with 14.9 Gb size, though it has not yet been made public [11].

Several attempts have been made to construct onion genetic linkage maps and develop trait-linked genetic markers using the amplified fragment length polymorphism (AFLP), random amplified polymorphic DNA (RAPD), restriction fragment length polymorphism (RFLP), simple sequence repeat (SSR), and cleaved amplified polymorphic sequence (CAPS) systems [12,13,14,15,16,17]. However, the density markers of these linkage maps were mostly low and inadequate for the fine mapping of the loci governing functional traits. In order to facilitate the rapid and accurate genetic mapping of complex traits, the evolving high-throughput modern sequencing technologies are providing better opportunities and platforms to the researchers and breeders. Single nucleotide polymorphisms (SNPs) are the frequently used molecular markers, preferred for their abundant presence, uniform distribution across the genome, and robust connotation to the traits of interest [18].

New software and modern tools in next-generation sequencing (NGS) technologies permit the cost-effective identification, assessment, evaluation and validation of SNPs at a large scale. Among the NGS approaches, genotyping by sequencing (GBS) is attractive to researchers for its cost-effectiveness and wider application for high-throughput analysis [19,20]. The GBS method can decrease the complexity of the genome by binding a barcode adapter to DNA and processing it with restriction enzymes for sequencing [21]. GBS technology works well either in the presence or absence of a reference genome sequence, and it has the advantage of being able to perform SNP discovery and genotyping concurrently [21]. GBS has been applied to a variety of plant genetics studies, including linkage mapping, QTL mapping, genome-wide association studies, genome selection, and genome diversity studies [21,22,23,24]. Furthermore, by selecting the appropriate restriction enzymes (REs), GBS can provide high SNP coverage in the gene-rich regions of the genome [25]. A large number of SNPs have been generated through GBS analysis to develop SNP markers and to construct genetic linkage maps for plant genetics and breeding [22,23,24,26]. GBS-based SNPs have been used to construct a genetic linkage map of onion and validated through Fluidigm-based markers [2]. The study reported a high-resolution genetic map consisting of eight linkage groups with a total genetic distance of 1339.5 cM [2]. Furthermore, GBS-SNPs have been used for the genetic mapping of QTLs controlling anthocyanin content (*qAC4.1* and *qAC4.2*) and synthesis (*qAS7.1*) on onion chromosomes 4 and 7, respectively [27].

In the present study, we constructed an onion genetic linkage map using the reference-free GBS de novo approach and resulting SNPs to identify QTLs controlling sugar content in an F_2_ onion population.

## 2. Results

### 2.1. Sugar Content Analysis of Onion Samples

The concentrations of the four types of sugar content, including sucrose, glucose, fructose, and total sugar, in onion bulks of the F_2_ segregating population are presented in Table 1 and Figure 1. The sucrose, glucose, fructose, and total sugar were recorded from 3.7 to 22.3 mg g^−1^, 8.6 to 73.3 mg g^−1^, 9.1 to 51.3 mg g^−1^, and 26.5 to 135.2 mg g^−1^, respectively (Table 1). The average values of the four sugar contents were observed as 7.6 mg g^−1^, 33.4 mg g^−1^, 21.0 mg g^−1^, and 62.1 mg g^−1^ for the sucrose, glucose, fructose, and total sugar, respectively (Table 1 and Figure 1). These values showed continuous variation for the sugar content, attributing that these traits are quantitative and controlled by polygenes. The correlation analysis revealed that the total sugar content was positively correlated with each sugar, for instance glucose (*r* = 0.97), fructose (*r* = 0.93), and sucrose (*r* = 0.65). Glucose had a highly significant positive correlation with fructose (*r* = 0.87) and sucrose was positively correlated, though not higher in value, with glucose (*r* = 0.52) and fructose (*r* = 0.46) (Figure 1).

### 2.2. SNP Detection and Genotyping-by-Sequencing (GBS) Analysis

GBS analysis was carried out using 156 F_2_ onion plants for SNP discovery and genotype identification. Approximately, 1.3 billion total raw reads and 207.3 Gbp sequences were obtained from the Illumina HiSeq X pair-end read sequencing (Table 2). The GBS raw data was demultiplexed according to the 156 barcode sequences which were assigned to each individual. The demultiplexed sequences of the 156 samples were trimmed by eliminating barcodes, adaptors, and low-quality sequences. The average length of trimmed reads was obtained as 1.17 Gbp, interpreting 85.2% of the total raw data (Table 2). The trimmed reads were mapped to the reference gene set of bulb onion [28]. As a result, the total number of mapped reads was about 565,805,698 where only 41.2% of the raw reads were mapped. The number of mapped regions and average depth of mapped regions were 1638,943 and 147.77, respectively (Table 2). The average length of the mapped regions was 1680.157 bp, which covered 0.77% of the onion reference gene set (Table 2). Using the information of the mapped data of 156 samples, a SNP matrix consisting of a total of 24,341 genome-wide SNPs was identified (Table 2). After filtering with a minimum depth of three, less than 30% missing proportion, and over 5% major allele frequency, finally, 815 SNPs were obtained (Table 2 and Appendix A).

### 2.3. Construction of an Onion Genetic Linkage Map

An onion genetic linkage map was constructed using 156 onion-segregating F_2_ plants and 216 GBS-based SNPs. The genetic map construction yielded a total of eight linkage groups (equal in numbers to the onion chromosomes), spanning into the genetic length of 827.0 cM (Table 3 and Figure 2). The number of markers on each linkage group ranged between 17 to 39 where an average number of markers and interval was 27 markers and 3.8 cM, respectively (Table 3 and Figure 2). The highest number of SNP markers was mapped on the linkage group 2 (Chr.2), whereas the lowest was on linkage group 4 (Chr.4) (Table 3 and Figure 2). Likewise, the linkage group 3 (Chr.3) consisted of SNPs positioned with the highest map length of 152.7 cM, while linkage group 4 (Chr.4) had the lowest genetic map length of 88.8 cM (Table 3 and Figure 2).

### 2.4. Comparison of the ‘16P118 × Sweet Green’ and ‘SP3B × H6’ Onion Genetic Linkage Maps

The ‘16P118 × Sweet Green’ genetic linkage map constructed in this study was compared with the previously reported onion genetic map ‘SP3B × H6’ to allocate the linkage groups to the corresponding chromosomes. Initially, through a BLAST search with the mapped transcript sequences, 43 common transcripts were identified (Figure 3). After that, the chromosomal location of the common transcript was compared, and chromosome numbers were assigned to the onion genetic map constructed in this study. These results showed that the eight linkage groups correspond to the respective chromosomes of the onion (Figure 3).

### 2.5. Identification of QTLs for Sugar Content in Onion

QTL analysis was performed for the soluble sugars, including sucrose, glucose, fructose, and total sugar content using the composite interval mapping (CIM) method. As a result, a total of four QTLs were identified for each trait (sucrose, glucose, fructose, and total sugar) explaining phenotypic variation (R^2^), ranging from 6.07 to 16.87%. One QTL linked to the sucrose *qSC4.1* was identified on chromosome 4 (Table 4 and Figure 4). The QTL *qSC4.1* was detected at 31.8 cM with LOD scores of 5.37, explaining R^2^ values of 11.47%. The QTLs associated with glucose, fructose, and total sugar *qGC5.1*, *qFC5.1*, and *TSC5.1*, respectively, were identified on onion chromosome 5 (Table 4 and Figure 4). These QTLs, (*qGC5.1, qFC5.1,* and *qTSC5.1*) were located at 73.0, 70.4, and 69.3 cM with LOD scores of 4.63, 4.15, and 3.14 and explained phenotypic variation (R^2^) by 16.87%, 15.39%, and 6.07%, respectively (Table 4 and Figure 4). Interestingly, the three QTLs (*qGC5.1*, *qFC5.1*, and *qTSC5.1*) were localized in the same genetic area and shared the same top marker (T65005.1_1614) on chromosome 5 (Figure 4 and Figure 5). The positive additive effects on sugar content were derived from the Sweet Green genotype which was observed in QTLs. A GBS-based marker, T2701.3_435 was the closest flanking marker to the *qSC4.1*, and T65005.1_1614 was the closest marker to the *qGC5.1*, *qFC5.1*, and *qTSC5.1* (Figure 4 and Figure 5). Sucrose level was significantly higher in the individuals with paternal alleles compared to the individuals carrying homozygous maternal alleles (Figure 5), while the other sugar content was significantly lower in the individuals with maternal alleles compared to the individuals carrying homozygous paternal alleles (Figure 5).

## 3. Discussion

In the present study, we constructed a GBS-derived genetic linkage map and applied it to the QTL mapping for sugar content in onion. Until now, the reference genome of the bulb onion has not been available publicly [11]. However, we reported a reference-free GBS-SNP-based onion linkage map using 156 F_2_ individuals comprising eight linkage groups (corresponding to the onion chromosomes) and successfully mapped four QTLs associated with sugar content in bulb onion.

DNA markers are substantial tools for crop improvement [29,30,31]. The first genetic map of onion was constructed by using the F_2_ population (BYG15-23 × AC43) primarily based on co-dominant molecular markers [12]. Later on, several studies into the construction of onion genetic linkage map using NGS technology were reported [2,27,29,32,33]. An onion linkage map was developed using 104 high-resolution melting (HRM), 90 CAPS and 11 Indel markers derived from NGS data [32]. Another linkage map using transcriptome data was constructed by using 479 SNPs, which also included the 140 previously reported EST markers [33]. GBS has been progressively exploited in genetic and genomic studies in a broad array of crop plants. The first reference-free GBS-SNPs based linkage map of onion was constructed using 92 F_2_ individuals and yielded 202 high-quality SNPs which were validated by Fluidigm assay [2]. A recent report about the GBS-SNP-based onion linkage map added 284 GBS-SNPs in a previously reported onion genetic map [12,29]. Another contemporary study about the construction of GBS-SNP-based onion linkage map and QTL mapping was reported by Choi et al. [27]. In that study, a high- linkage map was constructed using 319 GBS-SNPs and 34 HRM spanning at 881.4 cM genetic length [27]. These previous results are comparable with our outcomes. We developed an onion linkage map with 216 GBS-SNPs covering the total genetic length of 827 cM. Furthermore, comparative studies with the previously reported genetic map SP3B × H6 [27] revealed that our SNPs were evenly distributed across the onion genome attributing the reliability of our linkage map.

The sugar is supposed to affect the sweetness in onion bulbs containing glucose as the major sugar component, followed by fructose and sucrose [34]. Therefore, we performed HPLC analysis to evaluate these three sugar contents in fresh bulb onion. HPLC analysis indicated that glucose and fructose concentrations were always higher compared with sucrose for both parental lines and offspring. These results are in accordance with the previous research in the *Allium* species [7,35,36]. In contrast, in the case of sucrose, previous reports had shown a weak positive or negative correlation between glucose and fructose [10,36,37]. This discrepancy might be due to a difference in the evaluation criteria as we analyzed the sugar content in fully mature bulb onions soon after harvesting, whereas other researchers evaluated during storage, making it possible that the sugar content had appeared to change because of storage conditions after harvest [10,36,37]. Another possible reason that the difference in correlation of onion sugar composition compared with previous reports is highly likely to be due to the use of a different genotype in our study. Both the parental genotypes (16P118 and Sweet Green) showed a significant difference for the sugar content (Table 1).

The frequency distribution of sugar content-related traits in onion F_2_ population indicated a normal distribution with a slight left-skewed pattern for sugar content, including sucrose, glucose, fructose, and total sugar. These results indicated that sugar content is a quantitative trait and is controlled by multiple genes with additive effects. Although several genetic linkage maps in onion crops have been reported, there are few reports about dissection and mapping of quantitative traits. Some of the QTLs for sweetness have been identified in bulb onion [10,36,38]. Nevertheless, those studies reported QTLs associated with sucrose and fructose, but no QTLs had been detected for glucose. Noticeably, those QTLs were mapped with a low-density linkage map with uncertain QTL locations and are hard to utilize in breeding programs. Havey et al. [36] showed QTL on linkage group D, which was significantly associated with sucrose, using the linkage map created by the King et al. [12] with the LOD and phenotypic variation (R^2^) for sucrose 3.45 and 28.7%, respectively. Conversely, the genetic map used to detect that QTL was low-density, constructed by restriction fragment length polymorphisms (RFLP) markers, where the linkage group was also partially divided [12,36]. Several studies have been conducted related to onion sugar content evaluation [34,37,39,40,41,42]. However, limited QTLs associated with onion sugar content have been identified. This is the first study in onion to construct a SNP-based onion genetic linkage map through GBS analysis and QTL analysis for sugar content. Therefore, these results may lay the foundation to develop molecular markers and accelerate breeding programs for sugar content related traits in onion.

## 4. Materials and Methods

### 4.1. Plant Materials

An onion inbred line ‘16P118’ containing low sugar content and an inbred line ‘Sweet Green’ with high sugar content were selected as parents to cross and develop a mapping population. The mapping population was generated by self-pollination of an F_1_ hybrid produced and a total of 156 F_2_ individuals were used to construct the onion genetic linkage map and identify QTLs for sugar content. The plants were cultivated in the open farm fields of Mokpo Experimental Station of the National Institute of Crop Science (Muan, Korea) from October 2019 to June 2020.

### 4.2. Assessment of Sugar Content

Sugar content was analyzed according to the method described by Choi et al. [43] with a slight modification. In detail, 10.0 g of fresh onion bulbs with bark removed, which were advanced from each F_2_ individual plant, were homogenized in 20.0 mL of distilled water with homogenizer (T25, IKA, Labortechnik, Germany). Afterwards, the samples were put into an ultrasonic bath (UIL-30040H, MyungSung, Siheung, Korea) for 30 min and then into a shaking incubator (VS-8480SR, Vision, Daejeon, Korea) for 30 min at 4 °C, 120 rpm followed by centrifugation for 20 min at 4 °C, 150,000 rpm. In the next step, the supernatant was filtered through a Sep-pak C_18_ (Waters, Milford, MA, USA) and 0.45 uM PVDF syringe filter (Whatman, Little Chalfont, UK) and the filtrate was analyzed with a high-performance liquid chromatography (HPLC) system (1260 Infinity II series, Agilent Technologies, Santa Clara, CA, USA). Sugar concentrations of each individual were determined from standard curves (R^2^ ≥ 0.998) for sucrose, glucose, and fructose. All analyses were performed in hexaplicate, and results were expressed as mg g^−1^ fresh weight (FW). Total sugar content was calculated by summing the sucrose, glucose, and fructose content. One onion bulk per genotype were tested in six replications and their means were presented.

### 4.3. DNA Extraction

Total genomic DNA was extracted from the young leaf tissues of F_2_ individuals using a cetyl trimethylammonium bromide (CTAB) method [44]. The DNA was dissolved in 50 µL of triple-distilled water (TDW) and treated with 0.1 µL of 10 mg∙mL^−1^ RNase solution (Bio Basic Canada Inc., Ontario, ON, Canada). The quality and quantity of DNA was assessed by agarose gel electrophoresis and NanoVue (GE Healthcare, Chicago, IL, USA).

### 4.4. Genotyping-by-Sequencing Analysis

A total of 156 individuals and two plants from each parental line (‘16P118’ and ‘Sweet Green’) were subjected to GBS analysis, performed by SEEDERS (a bioinformatics company, Daejeon, Korea). Prior to next-generation sequencing (NGS), the quality and quantity of the extracted gDNAs were confirmed using agarose gel electrophoresis. The GBS library was constructed in the following process: adaptor annealing, DNA double digestion with *PstI* and *MsPI*, adaptor ligation, sample pooling, DNA purification, and multiplexed PCR [27,45]. The pooled GBS library was sequenced with an Illumina HiSeq X (Illumina, Inc., San Diego, CA, USA) using the pair-end read method. Demultiplexing of the raw sequences into individual samples were performed using the barcode sequence, followed by adapter sequence removal and sequence quality trimming. Adapter trimming was performed using cutadapt v. 1.8.3 [46], and sequence quality trimming was performed using the DynamicTrim and LengthSort program of SolexaQA V.1.13 [47]. The reads were aligned to the onion reference gene set using the Burrows-Wheeler Aligner (BWA) program v. 0.6.1-r104 [48]. Raw SNPs which mapped reads were detected and consensus sequences were extracted using SAMtools v.0.1.16 [49]. An SNP matrix was constructed between the 160 samples and the raw SNPs using the SEEDERS script in-house [50] and the SNPs were classified into homozygous (SNP read depth ≥ 90%), heterozygous (40% ≤ SNP read depth ≤ 60%), and others (homozygous/heterozygous; could not be distinguished by type) through an SNP filtering step.

### 4.5. Genetic Linkage Construction

Genetic linkage maps were constructed using the JoinMap version 4.1 software (Kyazma B.V., Wageningen, The Netherlands). Only SNPs fitting with the 1:2:1 ratio of the χ^2^-test were used (Appendix A). A logarithm of odds (LOD) score of 3.0 was regarded as the threshold to determine the significant linkage between markers. The map distance was calculated using the Kosambi mapping function [51]. Skewed SNPs were excluded by Chi-square test (*p* < 0.001), and the markers displaying identical segregation or more than five missing data points were eliminated. MapChart version 2.2 software was used for the visualization of the final genetic linkage maps.

### 4.6. QTL Analysis

Windows QTL Cartographer version 2.5 software [52] with the composite interval mapping (CIM) method was used to identify the QTLs. The LOD threshold for significance level was defined as the 50th highest with a 1000 times permutation test. QTL analysis was carried out using four sets of phenotypic data: sugar (SC), glucose (GC), fructose (FC), and total sugar (TSC) content.

### 4.7. Statistical Analysis

Standard deviation values were calculated for all the entries to compare the sugar content values using Excel. The concentration values were expressed on the basis of fresh weight. Correlation analyses were preformed among the total sugar content, the content of each individual sugar, and the composition of each sugar. Correlation analysis was performed based on Pearson’s correlation coefficient using R studio ver. 4.0.3 program. The correlation plot was drawn using the ‘Performance Analytics’ of the R program [53].

## 5. Conclusions

In summary, we performed GBS analyses using 156 F_2_ onion plants and constructed a genetic linkage map with 216 SNPs, consisting of eight linkage groups and covering a genetic length of 827.0 cM with an average marker interval of 3.8 cM. Through QTL analysis, we identified four QTLs associated with sucrose, glucose, fructose, and total sugar content in onion. The map information of the transcripts and SNP markers from the present study will contribute to completing the onion reference genome, and the QTL information for sugar content will be useful for molecular marker development for marker-assisted selection (MAS).

## Figures and Tables

**Figure 1 plants-10-02267-f001:**
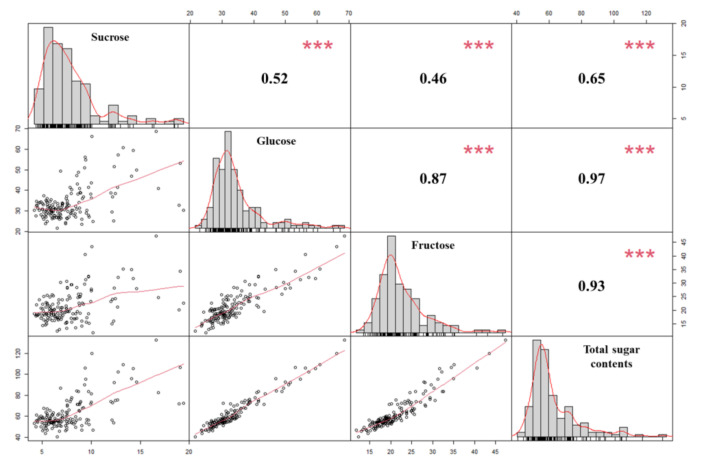
Overall correlation coefficients and frequency distributions among sucrose, glucose, fructose and total sugar content in 156 F_2_ onion mapping population. Red asterisk *** indicate the significance at the levels of *p* < 0.001.

**Figure 2 plants-10-02267-f002:**
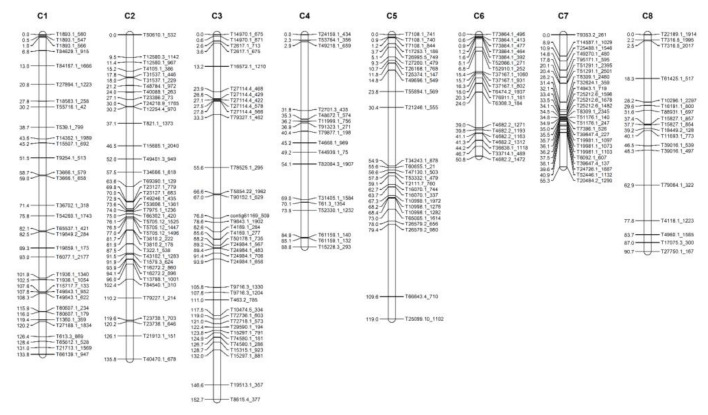
Genetic map of onion constructed by GBS-based SNP markers. Number on the left side corresponds to the genetic distance in cM from the top of each chromosome. Names on the right side indicate marker names.

**Figure 3 plants-10-02267-f003:**
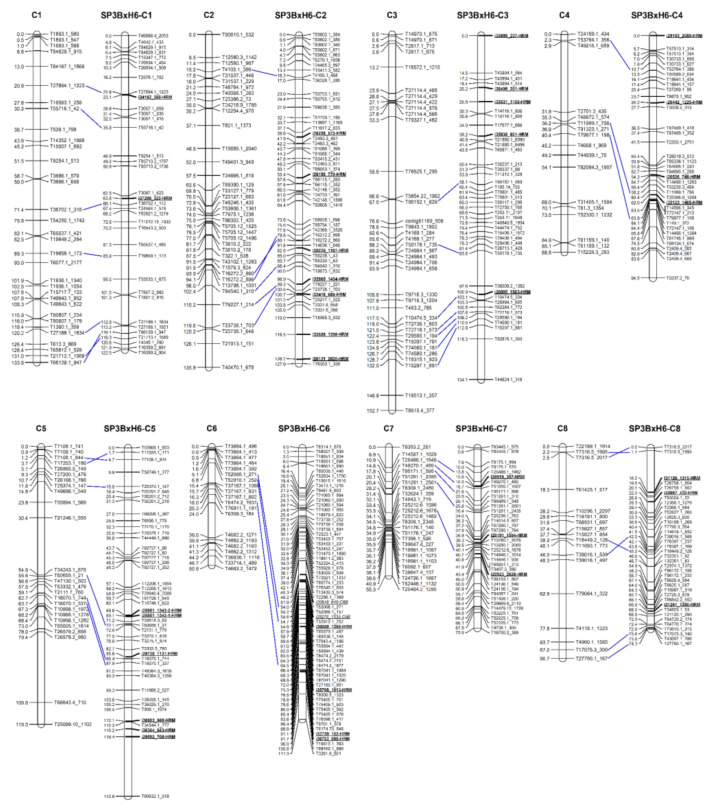
Comparison of two onion genetic linkage maps, the 16P118 × Sweet Green map constructed in this study and the SP3B × H6 map developed by Choi et al. [27]. Bar left or right number, map position (centi Morgan, cM); bar left or right name, marker name; dotted blue line, connection between the same transcript-based markers.

**Figure 4 plants-10-02267-f004:**
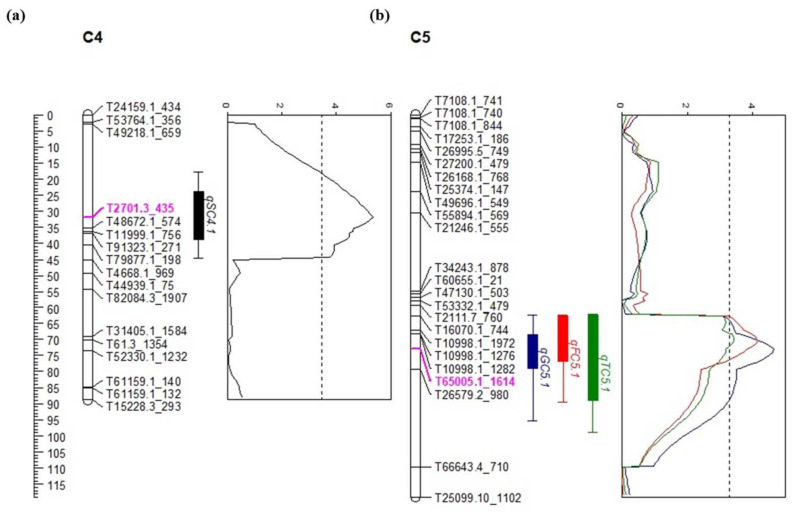
Quantitative trait loci (QTLs) associated with sugar content on chromosome 4 and 5. (**a**) QTLs position of sugar content (*qSC4.1*) on chromosome 4 of the onion genetic map. (**b**) QTLs position of glucose (*qGC5.1*), fructose (*qFC5.1*) and total sugar (*qTSC5.1*) content on chromosome 5 of the onion genetic maps. QTLs for *qSC4.1 qGC5.1, qFC5.1,* and *qTSC5.1* are represented by black, blue, red and green, respectively. QTL plot obtained by composite interval mapping (CIM) analysis.

**Figure 5 plants-10-02267-f005:**
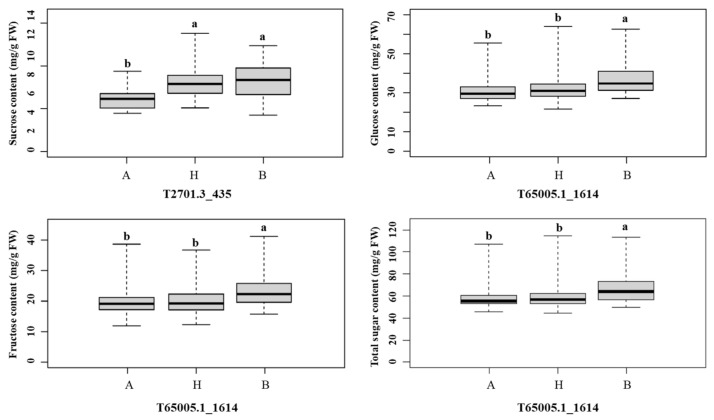
Phenotypic distribution of sugar content measured in the ‘16P118 × Sweet Green’ mapping population based on SNPs closet to the QTL peaks. Bars on graphs indicate standard error and different small letters (a and b) refer to significant differences (*p* < 0.05) according to Duncan multiple range test. A, genotype of female parent (16P118); B, genotype of male parent (Sweet Green); H, genotype of heterozygote.

**Table 1 plants-10-02267-t001:** Mean and range for sugar content in onion bulbs from 16P118 and Sweet Green and ranges of offspring from mapping population of 16P118 Ψ Sweet Green (mg g^−1^).

		Sucrose	Glucose	Fructose	Total Sugar
16P118 (female)		7.6 ± 0.4 a ^1^	31.9 ± 1.3 a	23.8 ± 1.3 a	63.2 ± 2.2 a
Sweet Green (male)		19.4 ± 3.3 b	30.4 ± 5.4 a	22.5 ± 3.7 a	72.3 ± 4.1 a
Offspring	Range	3.7 to 22.3	8.6 to 73.3	9.1 to 51.3	26.5 to 135.2
	Average	7.6	33.4	21.0	62.1

^1^ Significant differences (*p* < 0.05) according to Duncan multiple range test.

**Table 2 plants-10-02267-t002:** Summary of genotyping by sequencing data generated by using transcriptome sequences as a reference.

Summary of Illumina Sequencing	Data
Number of plants for multiplexing	156
Total number of raw reads generated	1,373,106,298
Total base number of raw reads (bp)	207,339,050,998 (207.3 Gbp)
Total number of demultiplexed reads	1,249,719,142 (91.0%)
Total number of trimmed reads	1,170,326,476 (85.2%)
Total number of mapped reads	565,805,698(41.2%)
Total number of mapped regions	1,638,943
Average depth of mapped region	147.77
Total length of mapped regions (bp)	1,680,157 (1.7 Mbp)
Total length of the reference gene set (bp)	280,586,290 (280.6 Mbp)
Coverage of the reference gene set	0.7676%
Total number of SNPs detected	24,341
Total number of SNPs filtered	815

**Table 3 plants-10-02267-t003:** Information of the onion genetic linkage map constructed in an F_2_ population of ‘16P118 × Sweet Green’.

Chromosome No.	Number of SNPs	Length of Linkage Distance (cM)
1	33	133.8
2	39	135.8
3	37	152.7
4	17	88.8
5	26	119.0
6	20	50.8
7	26	55.3
8	18	90.7
Total	216	827.0

**Table 4 plants-10-02267-t004:** Summary of QTLs for sugar content detected in an F_2_ population of ‘16P118 × Sweet Green’.

Trait	QTL Name	Chr.	Peak Position ^1^ (cM)	LOD ^2^	LOD Threshold ^3^	Additive Effect	Dominance Effect	R^2^ (%) ^4^
Sucrose	*qSC4.1*	4	31.8	5.37	3.48	1.15	0.78	11.47
Glucose	*qGC5.1*	5	73.0	4.63	3.52	3.12	−1.51	16.87
Fructose	*qFC5.1*	5	70.4	4.15	3.44	1.95	−1.21	15.39
Total sugar	*qTSC5.1*	5	69.3	3.41	3.40	4.84	−2.92	6.07

^1^ Positions of the markers on the linkage map. ^2^ LOD, logarithm of odds. ^3^ LOD threshold was determined with a 1000 times permutation test. ^4^ Percent of the phenotypic variation explained by the QTLs.

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
