# Peer review of "Genotyping-by-Sequencing Derived Genetic Linkage Map and Quantitative Trait Loci for Sugar Content in Onion (Allium cepa L.)"

_plants, 2021, doi:10.3390/plants10112267_

Round 1

Reviewer 1 Report

In this study, it was constructed SNP-based genetic linkage map of onion in 156 F2 onion plants at segregating population derived from a cross between the doubled haploid line '16P118' and inbred line 'Sweet Green' through genotyping-by-sequencing using an Illumina HiSeq X system.
The main objective of this work is to identify QTLs controlling sugar content in an F2 onion population
Unclear: "we identified the quantitative trait loci (QTLs) for the sucrose, glucose, fructose, and total sugar content across the onion genome."
The coding genes for sucrose, glucose, fructose synthesis are not present as complexes. It is not clear in what sense the quantitative trait loci may be in question.
NGS coverage of each genome of the line from F2 at segregating population was very low. As the authors collected for each SNP locus all the information among all the samples studied. As a result, no high-density linkage map was constructed, compared to if most of the genome had been sequenced.
The authors did not provide Genbank assay numbers for the sequenced genomes. 
No individual data for each line from F2 at segregating population.

Author Response

Thank you for your comments.

When we initiated this study, no proper onion reference genomes was available so, we used the onion transcriptome contig reported preciously. Therefore, it is difficult to detect the candidate genes related to sugar content when gene annotation was performed with the SNP information linked to the QTLs of present study.

The sequenced genome has been deposited to the NABIC (national agricultural biotechnology information center) in Korea and awaiting for the approval.

Finally, reflecting your opinion, I revised high-density linkage map to density linkage map.

Thank you.

Reviewer 2 Report

I would suggest that some modification of the approach used be made so as to achieve higher marker density of the map (eventually - supplementing SNPs with different marker types) as in the proposed manuscript the outcomes are very similar to a previously published research (ref. 2 from authors' list).

Author Response

Thank you for your comments.

When we started this study, we tried our vest to extract the maximum SNPs in the absence of an onion reference genome. However, the onion reference genome published in the Netherlands very recently in 2021. We will use that recent genome to improve our linkage map in further studies.

Thank you.

Reviewer 3 Report

The entire scientific work is valuable, with potential applied benefits, interesting for the reader, and based on current methods and approaches! The following minor remarks: Unify the spelling of Genotyping by sequencing or Genotyping-by-sequencing! After line 93, the decimal places are different in the text and the table corresponding to the text (Table 1). Line 359: The subtitle must be different from the subtitle at line 321! The entire text from line 116 to line 131 exactly repeats the information presented in Table 2. The authors must correct the text, or they must omit the table! In the paragraph after line 136, I think there should be a sentence describing why the genetic linkage map was constructed using 216 GBS-based SNPs, not all that you establish. The Conclusions section: The first sentence (line 343 - line 346) is exceptionally long, confused, and needs to be corrected!

Author Response

Thank you for your comments.

I commented on Line 323 to Line 324 why only 216 SNP was used. 

And the rest comments was revised to the manuscript.

Thank you.
